# Position: Creating High-Fidelity Synthetic Training Data Should Employ Multi-level Optimization

**Pengtao Xie** [1]  **Li Zhang** [1]  **Ruiyi Zhang** [1]

## Abstract

The reliance of machine learning (ML) models on large-scale, high-quality labeled training data incurs significant challenges in specialized domains where such data is expensive and difficult to obtain. A promising solution is the automatic creation of synthetic training data. However, current approaches — including data generation, automated annotation, and domain adaptation — often fail to explicitly use downstream model performance to guide the creation and refinement of synthetic training data. This position paper argues that multi-level optimization (MLO) is essential for producing high-fidelity synthetic data by enabling joint optimization of data generation, annotation, adaptation, and selection, all informed by downstream model performance. We advocate for MLO as a unified framework to address three critical challenges: (1) improving data generation by aligning synthetic data with model needs, particularly targeting class-specific deficiencies and worst-case robustness; (2) enhancing automated annotation through sequential verification and the use of large language models for more accurate labeling; and (3) enabling example-specific adaptation and selection to maximize data utility while preventing excessive over-adaptation. By facilitating end-to-end coordination across multiple learning stages, MLO offers a potential paradigm shift in synthetic data creation for data-scarce domains.

## 1. Introduction

Machine learning (ML), which aims at automatically extracting patterns from data and thereby making intelligent predictions, has demonstrated great success in numerous applications including autonomous driving, healthcare, smart cities, etc. The performance of ML hinges on the accessibility of large-scale, human-labeled training data. For example, the accuracy of image classification experienced a significant boost with the introduction of ImageNet (Deng et al., 2009) containing millions of labeled images. However, a large volume of data with high-quality human annotations is often very challenging to acquire in specialized domains (Schneider, 2014; Cao et al., 2019; Ibrahim et al., 2021) such as healthcare, structural biology, high-energy physics, legislation, etc. This challenge often stems from the high costs involved in obtaining high-grade human labels (Monarch, 2021) and data privacy concerns (Ibrahim et al., 2021). ML and deep learning (DL) models trained on small datasets often suffer significant problems such as poor generalization on test data (i.e., overfitting) (Zhang et al., 2021a), bias (Mehrabi et al., 2021), lack of robustness (Taori et al., 2020), etc.

Addressing data deficiency is essential for enabling the training of high-performance ML/DL models in domains with insufficient labeled data. A prominent strategy to overcome this challenge is the automatic creation of synthetic training data. Three main paradigms have been explored to this end. The first involves developing data generation methods (Ho et al., 2020; Song et al., 2021; Goodfellow et al., 2014; Kingma & Welling, 2014; Rezende & Mohamed, 2015) to produce high-fidelity synthetic labeled data, effectively supplementing the limited real training data. The second paradigm is automatic annotation of unlabeled data (Yu et al., 2015; Ratner et al., 2017; Varma & Ré, 2018; Pham et al., 2021). While labeled data is difficult to obtain, unlabeled data (sharing the same distribution as labeled data) is often abundant. If it can be automatically and accurately annotated, substantial amounts of labeled data can be yielded. The third paradigm is to adapt (Long et al., 2015; Ganin et al., 2016; Tzeng et al., 2017) and select (Ruder & Plank, 2017; Ge & Yu, 2017; Miao & Sankaran, 2022) labeled data from auxiliary sources for training models within a target domain.

Developing the three paradigms of methods is faced with three significant challenges respectively. First, how can we guarantee that generated data is effective for improving

[1]University of California San Diego. Correspondence to: Pengtao Xie <p1xie@ucsd.edu>.

*Proceedings of the $43^{rd}$ International Conference on Machine Learning*, Seoul, South Korea. PMLR 306, 2026. Copyright 2026 by the author(s).

downstream models? Second, how can we ensure that automatically annotated labels are accurate? Third, based on their closeness to a target domain $T$, some source examples need to be adapted into $T$ while others can be directly selected into $T$ (Blitzer et al., 2011; Tuia et al., 2016; Peng et al., 2017; Luo et al., 2019). How can we automatically decide example-specific adaptation/selection actions?

Existing methods do not adequately address these challenges. First, existing data generation methods (Ho et al., 2020; Song et al., 2021; Rombach et al., 2022; Azizi et al., 2023; Trabucco et al., 2023; Zhou et al., 2023; Zheng et al., 2023; Goodfellow et al., 2014; Brock et al., 2018; Mariani et al., 2018; Kingma & Welling, 2014; Rezende & Mohamed, 2015) mostly follow a two-step process: generating synthetic data and training a downstream model with the generated data. The generation process typically lacks guidance from the downstream model's performance. As a result, generated data may not be effective for improving this model. Second, existing data annotation methods (Yu et al., 2015; Ratner et al., 2017; Varma & Ré, 2018; Wal et al., 2021; Pham et al., 2021) typically lack mechanisms of employing a sequence of verification steps to authenticate and improve the accuracy of generated labels end-to-end. This shortcoming often leads to annotation errors. Third, existing domain adaptation methods (Long et al., 2015; Ganin et al., 2016; Tzeng et al., 2017; Kang et al., 2019; Hoyer et al., 2023) often unnecessarily adapt source data that is already within the target domain $T$, leading such data to potentially drift out of $T$. Existing source data selection methods (Ruder & Plank, 2017; Ge & Yu, 2017; Wang et al., 2019; Miao & Sankaran, 2022) often discard data that has not been chosen, despite its potential utility in training target models if suitably adapted.

To address the three key challenges and limitations of existing approaches, **this position paper argues that multi-level optimization is essential for creating high-fidelity synthetic training data by enabling end-to-end data generation, annotation, adaptation, and selection guided by downstream model performance.** Each of these challenges inherently requires executing multiple learning stages in a tightly integrated, end-to-end manner. For example, using downstream model performance to guide the generation of complex, labeled synthetic data involves a multi-stage pipeline: 1) training a deep generative model (DGM) (Ho et al., 2020; Goodfellow et al., 2014) to produce structured, complex labels; 2) training a conditional DGM to generate input data conditioned on those labels; 3) using the trained DGMs to create labeled data and training a downstream model $M$; and 4) validating $M$ and leveraging its performance as feedback signals to refine the DGMs for improved data generation. Existing methods often struggle to perform these stages in a fully integrated way, which limits the extent to which performance feedback from stage 4 can be used to

improve stages 1 and 2. This disconnect limits the ability to tailor upstream processes to downstream needs. While self-training methods (Xie et al., 2020; Amini et al., 2025) incorporate downstream feedback, they do not support end-to-end optimization across interdependent stages and lack mechanisms to use validation loss to guide data annotation. These methods typically rely on heuristic labeling strategies and do not offer a unified optimization framework that aligns data annotation with downstream model performance.

Multi-level optimization (Migdalas et al., 1998; Sato et al., 2021; Xie & Du, 2022), a relatively under-explored optimization framework, offers a principled solution to this challenge. In MLO, a series of nested optimization problems are structured hierarchically, where the solution to each lower-level problem becomes part of the objective function or constraints of its upper-level counterpart. This dependency structure naturally models the sequential and interdependent nature of multi-stage learning pipelines. To enable end-to-end optimization across $n$ stages, one can construct an MLO formulation with $n$ nested levels, where each level corresponds to a specific learning stage. The relationships and dependencies among stages are encoded in the nesting structure of the optimization problems. Solving them jointly ensures that information flows across the entire pipeline, allowing upstream stages to be optimized with direct feedback from downstream performance.

By capitalizing on the capabilities of MLO, we can achieve goals that are difficult for modular approaches. Specifically, we can construct a unified pipeline that integrates multiple learning stages — such as data generation, annotation, adaptation, and selection — within a single MLO formulation. This allows each stage to be informed by downstream model performance, ensuring that the resulting data directly contributes to improving the model's effectiveness.

In summary, this paper advocates the position that multi-level optimization offers a necessary and unifying framework for end-to-end data creation guided by downstream model performance. Traditional pipelines typically treat upstream data-centric processes as isolated modules, each optimized independently of the downstream model. This modularity hinders the ability to align data curation with model needs. In contrast, MLO supports feedback-driven learning, enabling downstream performance to directly inform upstream decisions. By jointly optimizing interdependent components in a hierarchical manner, MLO allows improvements in one stage to propagate throughout the pipeline.

**Conflict of Interest Disclosure**   There are no conflict of interest.

## 2. Multi-level Optimization

Multilevel optimization (MLO) (Migdalas et al., 1998) consists of a nested hierarchy of optimization problems, where optimized variables from lower levels influence the objective functions of upper levels, while non-optimized variables from upper levels are used to define the objective functions of lower levels. Below is a simple example of an $n$-level MLO problem:

$$
\begin{aligned}
&\min_{v_n} \ L_n \left(v_{n-1}^*(v_n)\right) && \text{Level } n \\
&\quad \ddots \\
&s.t. \ v_k^*(v_n) = \operatorname{argmin}_{v_k} \ L_k(v_k, v_{k-1}^*(v_n)) && \text{Level } k \\
&\qquad \ddots \\
&\ \ s.t. \ v_2^*(v_n) = \operatorname{argmin}_{v_2} \ L_2(v_2, v_1^*(v_n)) && \text{Level } 2 \\
&\ \ s.t. \ v_1^*(v_n) = \operatorname{argmin}_{v_1} \ L_1(v_1, v_n) && \text{Level } 1
\end{aligned}
\tag{1}
$$

The optimized variables $v_{k-1}^*$ from level $k-1$ are used to define the loss function $L_k$ at level $k$, while the non-optimized variables $v_n$ from level $n$ are utilized to define the loss function $L_1$ at level 1. This MLO formulation enables end-to-end execution of $n$ interdependent stages in an ML pipeline, with each level corresponding to a stage. By jointly solving the $n$ levels of optimization problems, the $n$ stages are seamlessly integrated and performed in a unified manner.

Bi-level optimization (Ghadimi & Wang, 2018; Dempe, 2020; Grazzi et al., 2020; Ji et al., 2021; Liu et al., 2021; Chen et al., 2023), which encompasses two levels of nested optimization problems, has been applied for neural architecture search (Liu et al., 2019; Zhang et al., 2021b), meta learning (Finn et al., 2017; Rajeswaran et al., 2019), hyperparameter tuning (Franceschi et al., 2018; Lorraine et al., 2020), etc. MLO (Vicente & Calamai, 1994; Migdalas et al., 2013; Sato et al., 2021; Raghu et al., 2021) with more than two levels has been applied for knowledge distillation (Xie & Du, 2022), meta pretraining (Raghu et al., 2021), self-supervised learning (He et al., 2021), etc.

## 3. MLO for End-to-End Generation of Training Data

The feasibility of utilizing generated data to improve model training has been demonstrated in several works (Azizi et al., 2023; Trabucco et al., 2023; Zhou et al., 2023). Many deep generative models have been developed for data generation (or augmentation) (Azizi et al., 2023; He et al., 2022; Besnier et al., 2020), including diffusion models (Sohl-Dickstein et al., 2015; Ho et al., 2020; Song et al., 2021), generative adversarial networks (Goodfellow et al., 2014; Mirza & Osindero, 2014), variational autoencoders (Kingma & Welling, 2014; Wu & Deng, 2023), normalizing flows (Tabak & Vanden-Eijnden, 2010; Rezende & Mohamed, 2015), GPT3 (Brown et al., 2020a), etc. However, existing methods separate data generation from downstream model training, which can result in generated data that is not effective for training downstream models. To address this issue, it is crucial to integrate data generation and downstream model training in an end-to-end manner, allowing the performance of the downstream model to guide the data generation process. In this section, we illustrate how to leverage MLO to achieve this goal. Furthermore, we illustrate how MLO can be used to develop advanced data generation strategies by leveraging class-specific (Gala & Xie, 2024) and worst-case performance.

The methods and empirical results in Subsections 3.1–5.2 are presented as illustrative case studies that support the central position of this paper. They demonstrate how multi-level optimization (MLO) can be instantiated across diverse scenarios to generate high-fidelity synthetic training data, highlighting its generality and role as a unifying framework. These methods are not intended as standalone technical contributions, but as concrete realizations of the broader MLO paradigm.

### 3.1. MLO for Generating Data with Complex Labels

In many ML applications, data labels often have a complex structure. For example, in semantic segmentation (Chen et al., 2017a), the output labels - segmentation maps - are represented as 2D matrices. Generating data with such complex labels is generally more challenging than generating data with simpler labels, such as discrete labels in classification tasks. In this section, we illustrate how MLO can be applied to generate data with complex labels. We construct an MLO-based framework consisting of 1) a Label Generation Model (LGM) which generates labels exhibiting complex structures, 2) a Data Generation Model (DGM) generating input data examples conditioned on generated labels (e.g., generating radiographic images (Chen et al., 2017b) from electromagnetic structures), and 3) a main model $W$ designed to predict labels from input data. Given a real dataset $D_{tr} = \{(x_j, y_j)\}_{j=1}^N$ used for training $W$ where $x_j$ is an input data example and $y_j$ is the output label annotated by humans, we can utilize the collection of labels $Y = \{y_j\}_{j=1}^N$ to train the LGM. To train the DGM, we construct a dataset $D_{dg}$ by reversing input-output pairs from the labeled training dataset $D_{tr}$. For each pair $(x, y) \in D_{tr}$, we swap $x$ and $y$, yielding a new pair $(y, x)$, and include it in $D_{dg}$. Since each $x$ is associated with a single $y$, this mapping is one-to-one.

The framework consists of four stages that are optimized end to end. **Stage I (Label Generation).** In this stage, the weights of a label generation model (LGM), denoted by $G_l$, are trained by minimizing a generative modeling loss $L_l$ on labels $Y$. For example, in denoising diffusion models, $L_l$ can be the negative evidence lower bound (Song et al.,

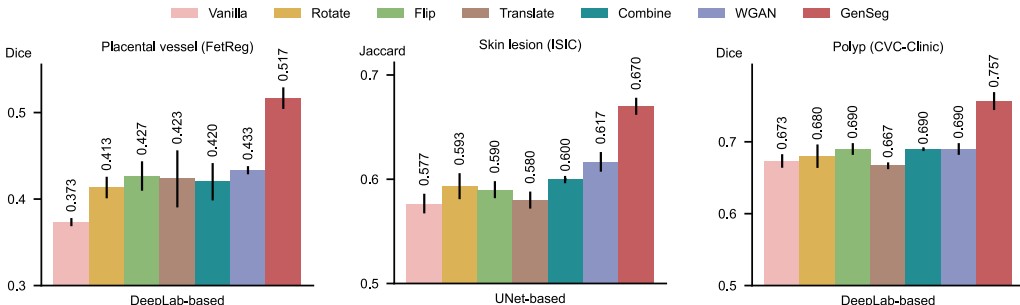

*Figure 1.* **GenSeg significantly outperforms widely used data augmentation and generation methods**. GenSeg's test performance compared to baseline methods including Rotate, Flip, Translate, Combine, and WGAN (Arjovsky et al., 2017), when used with UNet (Ronneberger et al., 2015) or DeepLab (Chen et al., 2017a) in segmenting placental vessels, skin lesions, polyps, intraretinal cystoid fluids, foot ulcers, and breast cancer using the FetReg (Bano et al., 2021), ISIC (Codella et al., 2019), CVC-Clinic (Bernal et al., 2015), ICFluid (Ahmed et al., 2022), FUSeg (Wang et al., 2020), and BUID datasets (Al-Dhabyani et al., 2020). The performance metrics are Dice score and Jaccard index. Please see Section E.3 for their definitions.

2021). The hyperparameters of the LGM, denoted by $H_l$ (e.g., neural architecture (Liu et al., 2019)), are fixed at this stage and will be optimized later. This stage corresponds to solving $G_l^*(H_l) = \arg\min_{G_l} L_l(G_l, H_l, Y)$. We explicitly write $G_l^*(H_l)$ to indicate that the optimal weights depend on the hyperparameters $H_l$, since $L_l$ itself is a function of $H_l$. **Stage II (Data Generation).** In this stage, the weights of a data generation model (DGM), denoted by $G_d$, are trained by minimizing a conditional generation loss $L_d$ (Saharia et al., 2022; Mirza & Osindero, 2014) on a dataset $D_{dg}$. As in Stage I, the hyperparameters $H_d$ of the DGM are provisionally fixed during this stage. This stage learns a conditional generator that maps labels to input data. **Stage III (Synthetic Data Generation and Model Training).** Using the trained LGM and DGM from Stages I and II, labeled synthetic data are generated. Specifically, the LGM $G_l^*(H_l)$ first produces a label $\hat{y}(G_l^*(H_l))$, which is then passed to the DGM $G_d^*(H_d)$ to generate an input example $\hat{x}(G_d^*(H_d), \hat{y}(G_l^*(H_l)))$. Repeating this process yields a labeled synthetic dataset $\widehat{D}(G_d^*(H_d), G_l^*(H_l))$. The main model $W$ is then trained by minimizing its task-specific loss $L_m$ (e.g., mean relative error for electromagnetic structure prediction (Chen et al., 2017b)) on the union of the synthetic dataset $\widehat{D}$ and real training data $D_{tr}$. **Stage IV (Validation and Hyperparameter Optimization).** The trained main model, denoted by $W^*(H_d, H_l)$ to emphasize its dependence on the generator hyperparameters, is evaluated on a human-labeled validation set $D_{val}$. The resulting validation loss $L_m(W^*(H_d, H_l), D_{val})$ serves as a measure of the quality of the generated data. If the synthetic data are low quality, a model trained on them will perform poorly on $D_{val}$. Therefore, the hyperparameters of the generation models, $H_l$ and $H_d$, can be optimized by minimizing the validation loss $L_m(W^*(H_d, H_l), D_{val})$, thereby encouraging the LGM and DGM to generate high-fidelity data that directly improves downstream model performance. The four optimization problems can be integrated into an MLO formulation:

$$
\begin{aligned}
\min_{H_d, H_l} \ & L_m\left(W^*(H_d, H_l), D_{val}\right) \\
s.t. \ & W^*(H_d, H_l) = \operatorname{argmin}_W \ L_m\left(W, \widehat{D}\left(G_d^*(H_d), G_l^*(H_l)\right)\right) \\
& \qquad\qquad\qquad + \lambda L_m\left(W, D_{tr}\right) \\
& s.t. \ G_d^*(H_d) = \operatorname{argmin}_{G_d} \ L_d(G_d, H_d, D_{dg}) \\
& \quad s.t. \ G_l^*(H_l) = \operatorname{argmin}_{G_l} \ L_l(G_l, H_l, Y)
\end{aligned}
$$
(2)

In lower levels, hyperparameters are fixed while optimizing lower-level weight parameters. These are then passed to upper levels, where validation loss is used to update the hyperparameters. The updated hyperparameters are fed back into the lower levels, and the process repeats until convergence. If the initial hyperparameter values are suboptimal, this iterative refinement allows for progressive recovery from poor initializations.

We conducted preliminary studies using the framework described in Eq.(2) to generate training data for medical image semantic segmentation models. Specifically, we applied this framework, referred to as GenSeg, across three segmentation tasks to produce training data for models such as DeepLab (Chen et al., 2017a) and UNet (Ronneberger et al., 2015). The main experimental settings are specified as follows. The Label Generation Model (LGM) consists of standard image augmentations (e.g., rotation, flipping), applied to human-labeled masks. The Data Generation Model (DGM) is a Pix2Pix-based mask-to-image GAN (Isola et al., 2017). The DGM's hyperparameters $H_d$ correspond to the learnable generator architecture (made differentiable via DARTS (Liu et al., 2019)). The weight parameters $G_d$ include generator and discriminator weights, trained via the standard GAN objective. The segmentation model (UNet or DeepLab) is trained with pixel-wise cross-entropy loss. Since the LGM has no learnable parameters, the first level in the four-level MLO formulation (Eq. 2) is omitted, yielding a three-level problem, solved using the Betty (Choe et al., 2023) framework. We evaluated GenSeg on FetReg,

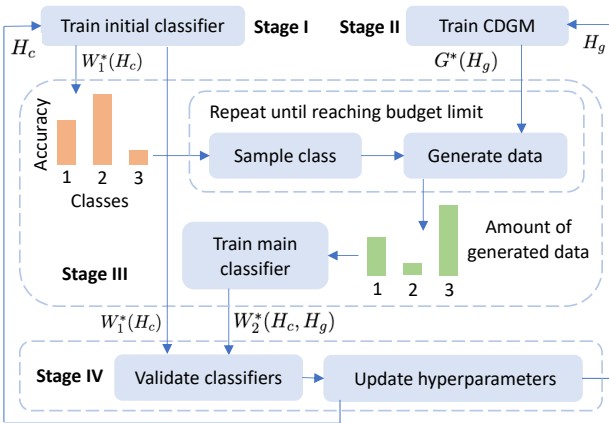

*Figure 2.* Class-specific performance guided data generation.

ISIC, and CVC-Clinic for placental vessel, skin lesion, and polyp segmentation, respectively. GenSeg was compared against standard augmentation methods (rotation, flipping, translation and combinations) and a WGAN-based generator (Arjovsky et al., 2017). For GenSeg, the training data was split into $D_{tr}$ and $D_{val}$ with a 4:1 ratio; baselines used the full dataset. As shown in Figure 1, GenSeg achieved significantly better performance compared to baselines. Unlike prior work, which separates data generation from downstream model training, our MLO framework uses downstream validation performance to guide data generation, resulting in task-specific and more effective training data. These findings highlight the benefits of end-to-end data generation leveraging MLO.

### 3.2. MLO for Leveraging Class-Specific Performance to Generate Data

Gala & Xie (2024) demonstrate how MLO can exploit class-specific performance for needs-based data generation. In practice, the volume of generated data is constrained by computational and memory limits, such as GPU capacity and training cost. Given a finite generation budget, it is therefore more effective to allocate data across classes based on their specific needs rather than generating data uniformly. Classification models typically perform better on some classes than others (Huang et al., 2016; Kim et al., 2020), and underperforming classes generally require more training data. Evaluating class-specific performance and using it to guide data generation enables more efficient use of limited resources.

To achieve this goal, Gala & Xie (2024) constructed an MLO-based framework with four stages (Figure 2). In stage I, the weights $W_1$ of an initial classifier are trained to assess the performance of individual classes, while the classifier's hyperparameters $H_c$ are temporarily fixed. In stage II, the weights $G$ of a Class-to-Data Generation Model (CDGM)

are trained. The CDGM takes a class name as input and generates a data example that can be labeled by this class. The CDGM's hyperparameters $H_g$ are tentatively fixed in this stage. In stage III, the performance of the trained initial classifier $W_1^*(H_c)$ is evaluated for each class on a validation set $D^{(val)}$. Then the trained CDGM $G^*(H_g)$ generates data according to the performance of individual classes: more data is generated for lower-performing classes. Then the generated data is used to train a main classifier $W_2$. In stage IV, $W_2^*(H_c, H_g)$ and $W_1^*(H_c)$ are validated on $D^{(val)}$. $H_c$ and $H_g$ are updated by minimizing validation losses. These stages can be performed in an end-to-end manner by solving the MLO problem in Eq.(7) in (Gala & Xie, 2024).

Experimental results in (Gala & Xie, 2024) on imbalanced classification and neural architecture search demonstrated that guiding data generation based on class-specific performance - specifically, generating more training data for underperforming classes - significantly outperforms baselines that allocate the data generation budget based solely on class frequency, i.e., generating more data for less-frequent classes. In Appendix C, we demonstrate how MLO can be utilized to guide data generation based on worst-case performance.

## 4. MLO for End-to-End Annotation of Unlabeled Data

Another strategy to mitigate the lack of labeled data is to automatically annotate unlabeled data. Various methods (Yu et al., 2015; Ratner et al., 2016; 2017; Varma & Ré, 2018; Suchi et al., 2019; Das et al., 2020; Wal et al., 2021; Zhou et al., 2022; Wang et al., 2023) have been developed for this purpose, based on user-provided labeling functions (Ratner et al., 2017), auto-created labeling rules (Varma & Ré, 2018), human-in-the-loop labeling (Wal et al., 2021), etc. A critical issue in automated data annotation is ensuring the accuracy of generated labels. In manual annotation pipelines, human-annotated labels typically undergo a series of verification procedures (Willemink et al., 2020; Munro & Monarch, 2021; Braun, 2023). Labels that do not meet the requisite standards at each phase are re-annotated. Such a multi-step validation process is highly desirable in automated annotation methods as well. However, Existing methods (Yu et al., 2015; Ratner et al., 2017; Varma & Ré, 2018; Pham et al., 2021) lack this crucial mechanism, as they often treat data annotation and label verification as separate processes, potentially leading to inaccuracies in the resulting annotations. To address this issue, integrating data annotation and label verification into an end-to-end framework is essential, enabling verification performance to inform and refine the annotation process. In this section, we demonstrate how MLO can be utilized to achieve this integration. Furthermore, we demonstrate how MLO facilitates

the development of advanced data annotation strategies by utilizing large language models.

## 4.1. MLO for End-to-End Label Annotation and Verification

We construct an MLO-based end-to-end data annotation framework that sequentially verifies the accuracy of automatically generated labels and uses the verification outcomes to iteratively improve label quality. We consider two verification stages in the framework: 1) self-consistency verification, which checks for label consistency across data subsets; and 2) supervised verification, which assesses whether the auto-labeled dataset can enhance the generalization performance of a supervised learning task.

Specifically, the framework annotates a large unlabeled dataset $D_u$ given a small labeled dataset $D_l$. It comprises 1) an Annotation Network (AN) with weights $U$ and hyperparameters $A$, responsible for generating labels for $D_u$, 2) a Consistency Verification Network (CVN) $V_c$ evaluating the quality of labels produced by the AN based on a self-consistency criterion, and 3) a Supervised Verification Network (SVN) $V_s$ which checks the correctness of generated labels by assessing performance improvement on a supervised task. The framework involves four end-to-end stages. In Stage I, the AN assigns labels to each data point in $D_u$. The newly labeled dataset is then split into a training set $D_{tr}(U, A, D_u)$ and a validation set $D_{val}(U, A, D_u)$. The CVN is trained by minimizing the prediction loss $L$ on $D_{tr}(U, A, D_u)$. In Stage II, the trained CVN $V_c^*(U, A)$ is evaluated on $D_{val}(U, A, D_u)$. A lower evaluation loss, depending on $U$ and $A$, signifies that the generated labels are consistent across the two subsets. To encourage such consistency, $U$ is trained by minimizing this loss, with $A$ temporarily fixed. In Stage III, the AN $U^*(A)$ trained in stage II produces new labels for $D_u$, followed by training the SVN on the labeled dataset $D(U^*(A), D_u)$. In stage IV, the trained SVN $V_s^*(A)$ is validated on $D_l$. A high validation loss indicates inaccurate labels. This feedback is used to refine the hyperparameters $A$ of the Annotation Network, which are optimized to reduce validation loss and improve label quality. The four stages can be performed end-to-end by solving the four-level optimization problem provided in Eq.(4) in the appendix.

## 4.2. MLO for Harnessing Large Language Models to Annotate Data

Next, we demonstrate how multi-level optimization (MLO) can be used to harness large language models (LLMs) for data annotation. Both text-based LLMs such as GPT-3 (Brown et al., 2020b) and LLaMA (Touvron et al., 2023), as well as multimodal LLMs such as LLaVA (Liu et al., 2023) and MiniGPT-4 (Zhu et al., 2023), have shown strong zero-shot learning capabilities (Brown et al., 2020b). When provided with appropriate prompts, these models can extract rich semantic information from data and generate labels without human supervision. The key challenge is that the quality of the generated labels depends critically on the prompt. To address this, we construct an MLO-based framework that automatically learns prompts that guide an LLM to produce high-fidelity annotations. The framework consists of three components: a *Prompt Generation Model (PGM)* with weights $G$ and hyperparameters $A$, a large language model $M$, and a *Supervised Verification Network (SVN)* $V$. The PGM generates prompts, the LLM produces labels for unlabeled data based on these prompts, and the SVN evaluates the quality of the generated labels using downstream supervision. The framework is organized into three end-to-end stages. **Stage I (Prompt Generator Pretraining).** The PGM is first trained on a large collection of prompts and instructions from diverse sources. During this stage, the PGM's weights $G$ are optimized while its hyperparameters $A$ are fixed. This stage equips the PGM with the ability to generate syntactically and semantically meaningful prompts. **Stage II (Prompt-Based Annotation and Verification).** Given the trained PGM $G^*(A)$, a prompt is generated as $p(G^*(A))$. Since textual prompts are discrete and non-differentiable, Gumbel-Softmax (Jang et al., 2017) is used to enable gradient-based optimization. For each unlabeled example $x \in D_u$, the LLM $M$ takes $x$ together with the generated prompt as input and produces a label $l(M, p(G^*(A)), x)$. Applying this process to all unlabeled data yields an automatically labeled dataset $D_{al}(M, G^*(A), D_u)$. The SVN $V$ is then trained on this auto-labeled dataset to learn to predict labels in a supervised manner. **Stage III (Validation-Guided Optimization).** The trained SVN, denoted by $V^*(M, A)$, is evaluated on a human-labeled validation set. The validation performance reflects the quality of the labels produced by the LLM under the current prompts. This performance signal is used to update the LLM $M$ (via efficient low-rank adaptation (Hu et al., 2021)) and the PGM hyperparameters $A$, so as to maximize validation accuracy. Through this process, the PGM learns to generate prompts that guide the LLM to produce increasingly accurate annotations. These three stages are optimized end to end by solving the MLO problem defined in Eq. (5) in the appendix. Finally, while the above description focuses on textual prompts and LLMs, the framework is general and can accommodate other forms of prompts and pretrained models. Examples include visual or point-based prompts and foundation models such as CLIP (Radford et al., 2021), BLIP (Li et al., 2022), and the Segment Anything model (Kirillov et al., 2023), where prompts may take forms other than text.

# 5. MLO for End-to-End Adaptation and Selection of Source Data

One approach to mitigating limited labeled data in a target domain (e.g., a rural clinic) is to adapt and select data (Long et al., 2015; Tzeng et al., 2017; Hoyer et al., 2023; Ruder & Plank, 2017; Miao & Sankaran, 2022) from auxiliary sources (e.g., urban hospitals). Domain adaptation methods (Long et al., 2015; Tzeng et al., 2017; Hoyer et al., 2023) focus on transforming source examples into the target domain but often overlook domain overlap: adapting source examples that already lie in the target domain is unnecessary and may even shift them away from the target distribution. Conversely, source selection methods (Ruder & Plank, 2017; Miao & Sankaran, 2022) identify examples closely aligned with the target domain but typically discard unselected data that may still be informative. To address these limitations, example-specific adaptation and selection are needed (Xie et al., 2024). Xie et al. (2024) show how MLO can be used to jointly learn these decisions. Building on this framework, we further demonstrate how MLO enables advanced data adaptation by using simulation models to transform simulated data into real-world data.

## 5.1. MLO for Example-Specific Adaptation or Selection of Source Data

To address the lack of mechanisms for deciding whether source examples should be directly selected or adapted in settings with domain overlap, it is necessary to learn a Domain Distance Metric (DDM) that measures the proximity of each source example to the target domain and guides this decision. This requires end-to-end optimization of multiple interconnected stages, including DDM learning, source selection and adaptation, target model training, and validation, so that target validation performance directly informs the learning of the DDM. Xie et al. (2024) proposed an MLO-based framework to achieve this objective.

The framework consists of four stages executed end-to-end. In Stage I, a DDM network (DDMN) is trained via self-supervised learning (SSL). Since SSL labels are heuristically assigned and may be noisy, each SSL example is assigned a learnable weight that scales its loss, allowing noisy examples to be automatically down-weighted or eliminated. The trained DDMN is then used to predict whether each source example lies inside or outside the target domain. In Stage II, source examples identified as out-of-domain in Stage I are adapted into the target domain using the trained DDMN. In Stage III, the in-domain source data, adapted source data, and target training data are jointly used to train the target model. In Stage IV, the target model is evaluated on a validation set, and the SSL data weights from Stage I are updated by minimizing the validation loss. All stages are optimized end-to-end by solving the MLO problem defined in Eq. (8)

of (Xie et al., 2024).

Experiments in (Xie et al., 2024) on text classification and pathology imaging visual question answering show that example-specific adaptation or selection of source data using MLO significantly outperforms baselines that either adapt all source data to the target domain or select a subset of source data while discarding the remainder.

## 5.2. MLO for Leveraging Simulation Models to Adapt Simulated Data to Real Data

Next, we demonstrate how MLO can be applied to map simulated data onto the distribution of real data. In many applications (e.g., uncovering electromagnetic structures from radiographic images (Kochetkov et al., 2022)), there are well-established simulation models (Speiser et al., 2021; Chen et al., 2017b; Kochetkov et al., 2022) capable of synthesizing labeled data. However, adapting such synthetic data to real data presents a significant challenge. To tackle this, we can construct an MLO-based framework that capitalizes on the simulation model to perform reliable adaptation. The framework consists of five end-to-end stages. In stage I, the weights of a Domain Classification Network (DCN) are learned to distinguish simulated data from real data, with its hyperparameters tentatively fixed. In stage II, simulated input data is transformed into real data. For each simulated input example $X$, a small perturbation $\Delta_x$ is learned such that the perturbed input $X + \Delta_x$ is classified as real by the DCN trained in stage I. In stage III, the simulation model infers the label for $X + \Delta_x$. Specifically, a small perturbation $\Delta_y$ is added to the label $Y$ of $X$, and the perturbed label $Y + \Delta_y$ is fed into the simulation model which generates a new data example $S(Y + \Delta_y)$. $\Delta_y$ is optimized to maximize the closeness between $S(Y + \Delta_y)$ and $X + \Delta_x$. In stage IV, a main model is trained on the transformed data. In stage V, the trained main model is validated on real data and the DCN's hyperparameters are optimized by minimizing the validation loss. These stages can again be performed end-to-end by leveraging MLO.

# 6. Alternative Views

While this paper advocates end-to-end multi-level optimization (MLO) as a unified framework for creating synthetic training data, many prior works adopt a modular pipeline perspective, in which data generation, representation learning, and model adaptation are optimized as separate stages. This viewpoint emphasizes decomposition, reuse, and scalability, and has been highly influential across machine learning domains.

One prominent line of work treats synthetic data generation as an independent problem. Generative models such as GANs (Goodfellow et al., 2014; Shrivastava et al., 2017) and

diffusion models (Ho et al., 2020) are trained to approximate data distributions and are subsequently used to augment downstream training sets, especially in low-resource scenarios. In this view, improving the fidelity and diversity of generated samples is expected to translate into better downstream performance, without explicitly coupling generation to a specific task or model.

Another widely adopted alternative is self-supervised learning (SSL) (Chen et al., 2020; He et al., 2020; Azizi et al., 2021), which separates representation learning from task-specific supervision. SSL methods aim to learn general-purpose features from large amounts of unlabeled data using proxy objectives, under the assumption that such representations can be efficiently transferred to a variety of downstream tasks through fine-tuning. This two-stage paradigm has proven effective across vision, language, and multimodal settings, and is particularly attractive when labeled data is scarce but unlabeled data is abundant.

Transfer learning further reflects this modular philosophy. In computer vision, convolutional neural network backbones pretrained on ImageNet are routinely reused for downstream tasks such as object detection and segmentation (Girshick et al., 2014; He et al., 2016). In natural language processing, large pretrained language models such as BERT (Devlin et al., 2019) and GPT (Brown et al., 2020a) follow a similar paradigm: pretraining on large, general-purpose corpora followed by task-specific fine-tuning. These approaches assume that broad, task-agnostic pretraining captures transferable knowledge that can be adapted to many downstream applications with minimal additional supervision.

From this modular perspective, end-to-end coordination across stages is not strictly required. Each component is optimized according to its own objective, with the expectation that improvements at individual stages will cumulatively benefit downstream performance. This design prioritizes simplicity, interpretability, and the reuse of pretrained components, enabling modules to be developed and deployed independently. In contrast, multi-level optimization (MLO) can be viewed as an alternative formulation that introduces tighter coupling between stages when stronger alignment is needed, rather than a replacement for these established modular paradigms.

## 7. Discussions

While MLO introduces nested optimization problems, recent algorithmic and system-level advances have made such formulations increasingly tractable in practice. Efficient techniques such as first-order implicit differentiation (Choe et al., 2023) and hypergradient approximation (Liu et al., 2019) avoid explicit second-order computations while maintaining stable optimization, enabling MLO to scale across

domains. In NLP, tri-level MLO methods for text generation guided by downstream classification performance (Somayajula et al., 2022) outperform modular baselines on large-scale datasets such as CNN/DailyMail (approximately 300k examples), using models including BART-Large (406M parameters) and RoBERTa-Large (355M parameters). Class-specific, performance-guided data generation via tri-level optimization further improves over frequency-based baselines (Gala & Xie, 2024). In multimodal learning, tri-level optimization has been applied to select high-quality self-supervised examples for visual question answering (He et al., 2021), and more complex four-level formulations enable effective example-specific adaptation or selection of source data (Xie et al., 2024), with consistent gains across text and pathology imaging tasks. At the system level, frameworks such as Betty (Choe et al., 2023) automate gradient tracking and memory management, enabling MLO to scale to ImageNet-sized datasets with runtime comparable to non-MLO baselines. Consistent with prior work, our GenSeg framework for medical image segmentation (Section 3.1) demonstrates that MLO can be trained stably using existing toolkits and yields consistent improvements over modular approaches in ultra low-data settings. Overall, these results show that although MLO appears complex theoretically, recent algorithmic and system advances have made it practical at scale, offering improved alignment between data-centric processes and downstream performance. Please see Section A for a more detailed discussion.

## 8. Call to Action

To fully realize the potential of multi-level optimization (MLO) for high-fidelity synthetic data creation, coordinated efforts across research, systems, and application communities are required. We encourage researchers to develop practical MLO formulations that directly address data-centric challenges—such as data generation, annotation, adaptation, and selection—under data-scarce conditions, with particular attention to handling non-differentiability, long optimization chains, and noisy supervision, and to prioritizing evaluation in low-data regimes where MLO is most impactful. For algorithm and systems developers, reducing the engineering complexity of MLO remains essential; advances in scalable hypergradient computation, memory-efficient training, and robust system-level support will be critical to enabling broader adoption. For practitioners in data-scarce domains, including healthcare, scientific research, and low-resource languages, we advocate treating data creation as a first-class optimization problem, where downstream performance explicitly guides synthetic data generation, annotation, and adaptation. Finally, the broader ML community should adopt more holistic evaluation practices that account for how upstream data decisions influence downstream performance, robustness, and failure modes. Together, these efforts can en-

able principled, end-to-end data creation pipelines in which synthetic data is not only realistic but demonstrably useful. See Section B for further discussion.

## 9. Conclusions

This position paper highlights MLO as a critical framework for creating high-fidelity synthetic training data. Unlike existing methods that separate data generation, annotation, and adaptation from downstream model training, MLO integrates these processes end to end, allowing upstream tasks to be directly guided by downstream performance. This coordination leads to improved data efficiency, accuracy, and robustness, and enables advanced strategies such as class-specific data generation and worst-case performance optimization. Moreover, MLO naturally accommodates LLMs and simulation models, demonstrating its versatility in addressing data-scarcity challenges across domains.

## Acknowledgements

We acknowledge funding support from NSF IIS2405974, NSF IIS2339216, NIH R35GM157217, and NIH R21GM154171. We thank the reviewers for valuable feedback.

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

## A. Practicality and Scalability of Multi-Level Optimization

While MLO introduces nested optimization problems, recent algorithmic and system-level advances have shown that such formulations are increasingly tractable in practice. On the algorithmic side, efficient techniques for handling nested optimization—such as first-order implicit differentiation (Choe et al., 2023) and hypergradient approximation (Liu et al., 2019)—avoid explicit second-order computations while maintaining stable gradient flow. These approaches have enabled MLO to scale to large datasets and models across domains.

In NLP, tri-level MLO methods for text generation guided by downstream classification performance (Somayajula et al., 2022) outperform modular baselines on large-scale datasets such as CNN/DailyMail (approximately 300k examples), using large models including BART-Large (406M parameters) and RoBERTa-Large (355M parameters). Gala & Xie (2024) further demonstrate that class-specific, performance-guided data generation via tri-level optimization yields superior results compared to frequency-based baselines. Beyond text generation, MLO has also proven effective in multimodal learning: tri-level optimization has been used to select high-quality self-supervised examples for visual question answering (He et al., 2021), leading to consistent improvements over modular pipelines. More complex formulations have likewise been successful—example-specific adaptation or selection of source data via four-level optimization (Xie et al., 2024) significantly outperforms approaches that either adapt all data indiscriminately or discard unselected examples, with evaluations spanning both text classification and pathology imaging visual question answering.

At the system level, frameworks such as Betty (Choe et al., 2023) provide modular abstractions for multi-level optimization, automating gradient tracking, checkpointing, and memory management. Combined with algorithm–system co-design that avoids explicit second-order derivatives and leverages efficient distributed training, these tools enable MLO to scale to datasets as large as ImageNet (1M+ images) with runtimes comparable to non-MLO baselines. Consistent with these findings, our own experiments—such as the GenSeg framework for medical image segmentation (Section 3.1)—demonstrate that MLO can be trained stably using existing toolkits and optimization strategies, and that it consistently outperforms modular approaches in ultra low-data settings.

Beyond feasibility, MLO offers practical advantages over modular pipelines by enabling continuous feedback alignment between upstream and downstream components. This coordination reduces the trial-and-error cycles often required when tuning independent modules, potentially lowering overall development cost despite the added optimization complexity.

Taken together, these results indicate that although MLO formulations may appear complex theoretically, recent algorithmic and system-level developments have substantially reduced the barriers to their implementation. Both prior work and our experiments provide concrete evidence that MLO is feasible at scale and yields tangible benefits for data generation, adaptation, and downstream performance.

## B. Challenges and Solutions

### B.1. Challenges and Solutions in Developing MLO Methods for Data Generation

While diffusion models (Ho et al., 2020; Song et al., 2021) are highly effective for generating realistic data (Dhariwal & Nichol, 2021), their sampling procedures—based on iterative stochastic processes (Ho et al., 2020) or ODE solvers (Song et al., 2021)—are not end-to-end differentiable. This makes MLO formulations involving diffusion models incompatible with existing hypergradient-based optimization methods (Gudovskiy et al., 2021; Lorraine et al., 2020; Finn et al., 2017; Franceschi et al., 2018; Liu et al., 2019). To address this challenge, we introduce a policy hypergradient (PHG) approach based on reinforcement learning (RL) (Lillicrap et al., 2015). Given the parameter-passing graph of an MLO problem, we first convert it into a directed acyclic graph by removing edges associated with non-optimized parameters. Following a topological order, each optimization problem is approximated using a one-step gradient update (Liu et al., 2019). When the objective is non-differentiable, policy gradients are used instead, treating objective values as rewards. The resulting approximate solutions are propagated to downstream optimization problems. At the final level, where the objective is non-differentiable with respect to the diffusion model parameters, PHG is computed via the chain rule as a product of Jacobians. While most Jacobians are directly computable, the one associated with diffusion sampling is estimated using RL. Specifically, a diffusion model generates data, a downstream model is trained on this data, and validation performance is used as the reward signal to estimate the policy Jacobian. PHG descent is then applied to update the diffusion model parameters, and the process is repeated until convergence.

In addition, several alternative solutions are also available.

- **Gradient approximations**: Techniques such as the reparameterization trick (Jang et al., 2017) and score-function estimators (Song et al., 2021) enable gradient-based training even when parts of the generative process are non-differentiable.

- **Surrogate objectives**: For diffusion models, differentiable surrogate losses (e.g., based on intermediate denoising steps) can be optimized while still aligning with downstream task objectives.

- **RL-free alternatives**: Prior work has shown that reinforcement learning is not the only viable option. For example, Gala & Xie (2024) and Xie et al. (2024) leveraged differentiable surrogate formulations to successfully implement MLO-based data generation and selection, achieving performance gains without requiring full RL pipelines.

From a deployment perspective, this means that while naively coupling non-differentiable generative models with downstream tasks may be challenging, practical engineering solutions already exist and are actively being applied at scale. Our own experiments with GenSeg (Section 3.1) relied on differentiable surrogate losses, and we were able to train MLO stably without resorting to reinforcement learning. In summary, although generative models pose non-trivial challenges, the growing set of algorithmic techniques—gradient approximations, surrogate losses, and efficient toolkits (Choe et al., 2023)—makes their integration into MLO feasible and increasingly practical for real-world use cases.

### B.2. Challenges and Solutions in Developing MLO Methods for Label Annotation

Existing optimization algorithms (Gudovskiy et al., 2021; Lorraine et al., 2020; Finn et al., 2017; Franceschi et al., 2018; Liu et al., 2019) for MLO are limited to vanilla stochastic gradient descent, whereas LLMs are by default trained with more efficient adaptive optimizers (Kingma & Ba, 2014; Loshchilov & Hutter, 2019), such as Adam (Kingma & Ba, 2014). To integrate LLMs into MLO, it is imperative to resolve this incompatibility. To overcome this hurdle, we construct an MLO algorithm that is compatible with adaptive optimizers. The algorithm is based on implicit differentiation (ID) (Gudovskiy et al., 2021; Lorraine et al., 2020) and executes the following procedure iteratively at each level. Given the update rule $s_{l-1}$ of an adaptive optimizer (Kingma & Ba, 2014; Loshchilov & Hutter, 2019) at level $l-1$ and non-optimal parameters $v_l$ at level $l$, we can compute the derivative $\frac{\partial s_{l-1}}{\partial v_l}$ (needed for ID) using the chain rule as $\frac{\partial g_{l-1}}{\partial v_l} \frac{\partial s_{l-1}}{\partial g_{l-1}}$, where $g_{l-1} = \frac{\partial O_{l-1}}{\partial v_{l-1}^*}$, and $O_{l-1}$ and $v_{l-1}^*$ are the objective function and optimal parameters at level $l-1$. We can revise the adaptive optimizer such that $s_{l-1}$ only involves element-wise operations. Through this refinement, $\frac{\partial g_{l-1}}{\partial v_l}$ is a diagonal matrix. This can lower the computation/memory complexities from a cubic to a linear level. To circumvent the cubic complexity of Jacobian inversion, we can approximate the Jacobian with an identity matrix, which preconditions the original hypergradient.

### B.3. Challenges and Solutions in Developing MLO Methods for Source Data Adaptation and Selection

The utilization of the simulation model necessitates the development of a five-level MLO formulation, which results in computational and memory demands that exceed the capacity of a single machine. Existing MLO systems (Blondel et al., 2021; Deleu et al., 2019; Grefenstette et al., 2019; Choe et al., 2023), which are limited to data parallelism, struggle with scaling MLO to such a large number of levels. To overcome this hurdle, we can develop a new distributed system facilitating a hybrid of model and data parallelism. The system has a hierarchical architecture, which organizes machines into $K$ groups, each holding one data partition. Data parallelism is conducted between the groups, building upon our previous research on parameter servers (Xing et al., 2015; Zhang et al., 2017) and peer-to-peer broadcasting (Xie et al., 2018; 2016). Model parallelism is executed within each group. We can design a scheduler to parse an MLO graph into an execution graph where nodes represent the machines in a group and edges depict the communication between the machines. The scheduler automatically analyzes the compute/memory requirements of each optimization problem (OP) in the MLO graph, the volume of parameters exchanged between OPs during forward and backward passes, machines' hardware specifications and the communication bandwidth between them. Utilizing this information, graph matching (Yan et al., 2016) is performed between the MLO and execution graphs to ensure the optimal allocation of OPs to machines. We can design a load balancer to automatically determine the optimal number of unrolling steps for each OP to ensure an evenly distributed workload across the machines.

## C. MLO for Leveraging Worst-Case Performance to Generate Data

In this section, we demonstrate how MLO can be utilized to guide data generation based on worst-case performance (WCP). In many applications, ML models need to perform well not just on average-case test data, but also on worst-case challenging

*Table 1.* Comparison between GenSeg and the BBDM diffusion model on four medical image segmentation datasets including CVC, ICFluid, FUSeg, and ISIC.

| Method | CVC | ICFluid | FUSeg | ISIC |
|---|---|---|---|---|
| BBDM | $0.480 \pm 0.010$ | $0.684 \pm 0.006$ | $0.570 \pm 0.004$ | $0.638 \pm 0.009$ |
| GenSeg (ours) | $\mathbf{0.508} \pm 0.004$ | $\mathbf{0.697} \pm 0.010$ | $\mathbf{0.604} \pm 0.005$ | $\mathbf{0.673} \pm 0.011$ |

examples. We construct an MLO-based framework that generates data to enhance the worst-case robustness of a primary model $M$. This framework evaluates the WCP of $M$ and utilizes this information to direct the data generation process, producing examples that can be used to improve $M$'s performance under worst-case scenarios. This process involves a mini-max game (Goodfellow et al., 2014) between $M$ and a Worst-Case Evaluator (WCE) $E$, which generates validation data. Let $L_m(M, \widehat{D})$ denote the validation loss of $M$ on the data $\widehat{D}$ produced by $E$. The goal of $E$ is to create a worst-case validation dataset $\widehat{D}^*$ that maximizes $L_m$, implying that $L_m(M, \widehat{D}^*)$ would be the worst-case performance of $M$. This performance is then used to guide the generation of training data. A Training Data Generator (TDG) is employed to create data that trains an optimal main model $M^*$ to minimize $L_m(M, \widehat{D}^*)$. This amounts to solving $\min_M \max_{\widehat{D}} L_m(M, \widehat{D})$. The MLO-based method includes four end-to-end stages. In stage I and II, the weights $G$ and $E$ of the TDG and WCE are trained, while keeping their hyperparameters $H_g$ and $H_e$ temporarily fixed. In stage III, the trained TDG $G^*(H_g)$ generates data $\widehat{D}_t(G^*(H_g))$ for training the main model $M$ (together with real data $D_{tr}$). In stage IV, the trained WCE $E^*(H_e)$ generates validation data $\widehat{D}_v(E^*(H_e))$ to evaluate the trained main model $M^*(H_g)$. The WCE maximizes the validation loss $L_m(M^*(H_g), \widehat{D}_v(E^*(H_e)))$ w.r.t $H_e$ to generate the worst-case validation data. The TDG minimizes this loss w.r.t $H_g$ to generate training data for improving $M$'s worst-case performance. Moreover, to prevent the compromise of $M$'s average-case performance, the TDG will also minimize $M$'s validation loss on a human-labeled dataset $D_{val}$. These stages can be performed in an end-to-end manner by solving the following MLO problem:

$$
\begin{aligned}
&\min_{H_g} \max_{H_e} \; L_m(M^*(H_g), \widehat{D}_v(E^*(H_e))) + \lambda L_m(M^*(H_g), D_{val}) \\
&s.t. \; M^*(H_g) = \operatorname{argmin}_M \; L_m(M, \widehat{D}_t(G^*(H_g)) + \gamma L_m(M, D_{tr}) \\
&\quad\; s.t. \; E^*(H_e) = \operatorname{argmin}_E \; L_{wce}(E, H_e, D_{wce}) \\
&\qquad\; G^*(H_g) = \operatorname{argmin}_G \; L_{tdg}(G, H_g, D_{tgd})
\end{aligned}
\tag{3}
$$

Here, $D_{tgd}$ and $D_{wce}$ represent the datasets used to train the TGD and WCE models, respectively. $L_{tgd}$ and $L_{wce}$ denote their corresponding loss functions. $\gamma$ and $\lambda$ are tradeoff parameters.

## D. MLO Formulations

The MLO formulation for the framework described in Section 4.1 is as follows:

$$
\begin{aligned}
&\min_A \; L(V_s^*(A), D_l) \\
&s.t. \; V_s^*(A) = \operatorname{argmin}_{V_s} \; L(V_s, D(U^*(A), D_u)) \\
&\quad\; s.t. \; U^*(A) = \operatorname{argmin}_U \; L(V_c^*(U, A), D_{val}(U, A, D_u)) \\
&\qquad\; s.t. \; V_c^*(U, A) = \operatorname{argmin}_{V_c} \; L(V_c, D_{tr}(U, A, D_u))
\end{aligned}
\tag{4}
$$

The MLO formulation for the framework described in Section 4.2 is as follows:

$$
\begin{aligned}
&\min_{M,A} \; L_{svn}(V^*(M, A), D_{val}) \\
&s.t. \; V^*(M, A) = \operatorname{argmin}_V \; L_{svn}(V, D_{al}(M, G^*(A), D_u)) \\
&\quad\; s.t. \; G^*(A) = \operatorname{argmin}_G \; L_{pgm}(G, A, D_{pgm})
\end{aligned}
\tag{5}
$$

Here, $D_{pgm}$ represents the dataset used to train the PGM, $L_{pgm}$ denotes the loss function of the PGM, and $L_{svn}$ corresponds to the loss function of the SVN. $D_{val}$ refers to the human-labeled validation set.

## E. Additional Experiments

### E.1. Comparison to diffusion models

We compared GenSeg to a diffusion+finetuning baseline in semantic segmentation (Figure 1). We used the Brownian Bridge Diffusion Model (BBDM) (Li et al., 2023) to generate (image, mask) pairs, then trained a UNet segmentation model using

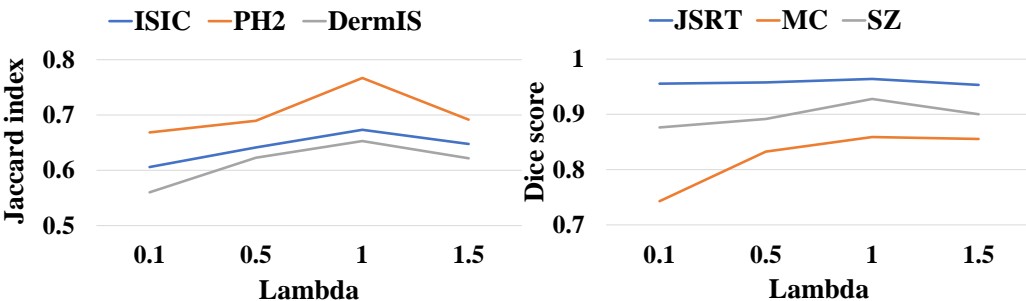

*Figure 3.* Ablation study results on how the tradeoff parameter $\lambda$ affects segmentation performance. ISIC: International Skin Imaging Collaboration (ISIC) dataset. JSRT: Japanese Society of Radiological Technology dataset. MC: Montgomery County chest X-ray dataset. SZ: Shenzhen chest X-ray dataset.

this data. Results on four datasets are shown in Table 1. GenSeg consistently outperforms BBDM. This highlights the benefit of using MLO to guide data generation based on downstream task performance, as opposed to decoupled generation and training.

### E.2. Hyperparameter sensitivity and tuning

We analyzed MLO's sensitivity to the hyperparameter $\lambda$ in Eq.(2) using the Japanese Society of Radiological Technology (JSRT) and International Skin Imaging Collaboration (ISIC) datasets. We varied $\lambda$ over $\{0.1, 0.5, 1.0, 1.5\}$. The results in Figure 3 show that MLO performance is stable across a broad range of $\lambda$ values.

Regarding hyperparameter tuning, we recommend a two-stage tuning strategy: (1) perform a coarse grid search to identify a stable range, and (2) refine via fine-grained search. The training data can be split into a training set, validation set, and a hyperparameter-tuning set. The tuning set guides final hyperparameter selection.

### E.3. Definition of Dice score and Jaccard index

The Dice score (Dice similarity coefficient) between a predicted segmentation $P$ and ground-truth segmentation $G$ is defined as:

$$\text{Dice}(P, G) = \frac{2|P \cap G|}{|P| + |G|}$$

where $|P|$ and $|G|$ denote the number of pixels in the predicted and ground-truth masks, respectively. The Jaccard index (Intersection over Union, IoU) is defined as:

$$\text{Jaccard}(P, G) = \frac{|P \cap G|}{|P \cup G|}$$

Both metrics are widely used in medical image segmentation to quantify the similarity between predicted and true regions.

## F. Additional Discussions

### F.1. Case Studies Demonstrating Multi-Level Optimization

The illustrative case studies throughout the paper support our overarching position by demonstrating the feasibility and versatility of MLO across key facets of the data pipeline:

- In Section 3, we show how MLO can guide data generation - not only for standard tasks, but also for class-specific needs and worst-case robustness. These examples highlight how MLO enables targeted, performance-aware data synthesis, rather than generating data in isolation.

- In Section 4, we apply MLO to automated data annotation, integrating sequential verification steps (e.g., consistency and supervised checks) and incorporating prompt generation for large language models. These demonstrate how annotation quality can be systematically improved by allowing verification performance to guide the annotation process.

- In Section 5, we show how MLO supports example-specific domain adaptation and selection, ensuring that data from auxiliary sources is used effectively by leveraging downstream model performance to guide data selection and adaptation.

## F.2. Design Logic for Feasible Multi-Level Optimization

Although multi-level optimization (MLO) introduces additional complexity compared to modular pipelines, recent advances in algorithms and systems make the proposed blueprint practically achievable. In this section, we clarify the design logic underlying each instantiation of MLO in this work and explain how feasibility is ensured through careful formulation and approximation.

**Data generation (Section 3).** For data generation, the key design choice is to treat the generative process as an upstream optimization problem whose objective is implicitly defined by downstream validation performance. Rather than requiring exact solutions to nested optimization problems, we rely on first-order approximations that enable stable end-to-end training. This strategy has been validated in recent work, including GenSeg (Section 3.1), where downstream segmentation performance effectively guides synthetic image generation, as well as class-specific data generation approaches (Gala & Xie, 2024) that selectively emphasize hard or underrepresented classes. These examples demonstrate that even approximate hypergradient signals are sufficient to align generative models with downstream objectives in practice.

**Annotation (Section 4).** For automated annotation, the framework decomposes the problem into sequential, interpretable stages—prompt generation, annotation, and verification—each of which can be incorporated into MLO with tractable approximations. Discrete components such as prompt selection are handled using continuous relaxations (e.g., Gumbel-Softmax (Jang et al., 2017)), while large language models are adapted using parameter-efficient techniques such as low-rank adaptation (Hu et al., 2021). Crucially, verification losses provide a stable supervisory signal that propagates through the annotation pipeline, ensuring that annotation quality improves in a manner consistent with downstream task performance rather than heuristic labeling accuracy alone.

**Adaptation and selection (Section 5).** For domain adaptation and data selection, feasibility is achieved by operating at the level of individual examples rather than global transformations. Example-specific adaptation and selection allow higher-level optimization problems to remain low-dimensional and well-conditioned. Prior work (Xie et al., 2024) has shown that four-level formulations of this kind can be trained stably and outperform approaches that indiscriminately adapt all data or discard unselected samples. Scalability is further enabled through algorithm–system co-design, where tools such as Betty (Choe et al., 2023) automate gradient tracking, checkpointing, and memory management, allowing MLO to scale to large datasets.

**General design principles.** Across all instantiations, several common principles ensure tractability: (1) higher-level optimization problems are simplified using first-order or implicit approximations rather than exact solutions; (2) proxy objectives at each stage are explicitly aligned with downstream validation loss to avoid optimization drift; and (3) parallelism and distributed execution are leveraged to mitigate the computational cost of nested training loops. Together, these choices reduce the gap between theoretical formulation and practical implementation.

In summary, while MLO is undeniably more complex than modular alternatives, the framework presented here is grounded in concrete design choices that have already been validated by recent empirical successes. By combining principled approximations with modern system support, the proposed blueprint demonstrates that MLO is not only conceptually appealing but also practically achievable with current algorithmic and computational resources.

## F.3. When Does Multi-Level Optimization Help? Data-Scarce vs. Data-Rich Regimes

It is important to clarify that the central argument of this work is not that multi-level optimization (MLO) improves model training in scenarios where ample labeled data is already available. Rather, we advocate MLO as a unifying framework for addressing data scarcity by enabling end-to-end coordination between upstream data-centric processes—such as data generation, annotation, and adaptation—and downstream model performance. In domains where labeled data is limited, simply training on the available data is often insufficient, and the creation of additional task-relevant synthetic or adapted data becomes a critical component of model development.

The key strength of MLO lies in its ability to maximize the utility of limited training data by using downstream performance

as an explicit guiding signal for upstream processes. When data is scarce, upstream modules trained in isolation are more likely to overfit proxy objectives or fail to address critical deficiencies such as rare classes or worst-case robustness. By tightly coupling upstream and downstream optimization, MLO selectively emphasizes the most beneficial synthetic or adapted examples, leading to substantial gains in low-data regimes. This setting is common in healthcare, scientific research, and low-resource languages, where data acquisition is expensive, constrained by privacy, or inherently limited.

When data is abundant, MLO can still provide benefits by aligning upstream processes with downstream objectives, but its relative advantage is less pronounced. Large real-world datasets often already provide broad coverage of class distributions and edge cases, enabling conventional ML/DL pipelines—such as generative augmentation and transfer learning—to achieve strong performance. In such high-resource settings, we do not position MLO as a replacement for standard training approaches. Instead, it may serve as a complementary framework that refines data utility or improves robustness, typically yielding more modest gains.

In practice, the suitability of MLO can be assessed using two indicators:

- **Performance plateauing due to insufficient labels** – If additional real data cannot be easily collected and models trained on existing data exhibit poor generalization (e.g., a large gap between training and validation or test performance), this suggests that synthetic or adapted data guided by MLO may be beneficial.

- **Domain-specific constraints** – In specialized domains such as healthcare, biology, or physics, where labeled data acquisition is costly or restricted, MLO is particularly well suited. In contrast, for tasks with abundant and inexpensive annotations (e.g., ImageNet-scale image classification), conventional pipelines already provide strong baselines, and MLO can be applied selectively to address rare classes or robustness challenges.

In summary, MLO should be viewed as a targeted solution for data-scarce domains, where feedback-driven coordination across the data pipeline is most impactful. In data-rich settings, simpler modular approaches often suffice, while MLO offers an optional layer of refinement that can be selectively deployed when additional alignment or robustness is desired.

### F.4. Multi-Stage Frameworks vs. Multi-Level Optimization

Conceptually, a multi-stage framework treats data generation, annotation, adaptation, and downstream model training as separate modules. Each stage is optimized independently with proxy objectives (e.g., maximizing likelihood in data generation, minimizing heuristic labeling errors in annotation), and the outputs are passed sequentially to the next stage. While this modular design is simple and widely adopted, it introduces several challenges:

- **Challenge 1**: Misalignment of objectives. Upstream modules are optimized without access to the true end goal of downstream generalization, which can result in synthetic or adapted data that looks realistic but is not task-relevant.

- **Challenge 2**: Lack of feedback across stages. Once an upstream stage has produced its output, downstream validation performance cannot inform corrections to the upstream process.

In contrast, multi-level optimization (MLO) explicitly encodes these interdependencies by formulating them as nested optimization problems. This offers several advantages:

- **Potential Solutions**: Validation performance from the downstream model is back-propagated to upstream processes, ensuring that synthetic data, annotations, and adaptation are directly tuned to improve the target task. Existing algorithmic advances (e.g., first-order implicit differentiation, distributed MLO frameworks such as Betty (Choe et al., 2023)) mitigate computational challenges.

- **Expected Outcomes**: As shown in our case study (GenSeg, Section 3.1) and prior work (Gala & Xie, 2024; Xie et al., 2024; Somayajula et al., 2022; He et al., 2021), MLO produces higher-fidelity synthetic data, improves worst-case and class-specific performance, and makes better use of auxiliary data. The result is improved downstream generalization, particularly in data-scarce domains where modular pipelines struggle.

In summary, while multi-stage frameworks optimize modules in isolation, MLO provides a principled and unified approach that aligns all stages with downstream goals.

### F.5. Downstream-Guided Multi-Level Optimization vs. Post-hoc Fine-Tuning

Our approach differs fundamentally from fine-tuning–based methods in terms of when and how downstream feedback is incorporated. Fine-tuning frameworks adopt a two-stage pipeline: first, the upstream component (e.g., data generator, annotator, or adaptor) is trained independently with heuristics or proxy objectives; only after this stage is completed is downstream model performance used to fine-tune parameters. This separation means that upstream modules are optimized without access to the true end-goal signal, and downstream performance influences them only post hoc.

By contrast, our formulation casts synthetic data creation as a multi-level optimization (MLO) problem, where upstream and downstream objectives are connected in a nested manner. At each level, the solution of a lower-level optimization problem (e.g., training a generator or annotator) is treated as a variable in the upper-level problem, which directly optimizes downstream validation performance. This structure has two major advantages:

- **Continuous, end-to-end feedback**: Rather than receiving downstream signals only after upstream training is complete, upstream modules are updated iteratively during their training based on downstream validation loss. This ensures that the data generation, annotation, or adaptation processes are progressively shaped to serve downstream model needs.

- **Joint optimization of interdependent stages**: Nested optimization captures dependencies across multiple stages (e.g., label generation → data generation → model training) and prevents misalignment that arises when these modules are trained in isolation. Improvements in one stage propagate through the pipeline, leading to more effective synthetic data tailored for the target task.

In short, fine-tuning treats downstream performance as an afterthought, while MLO uses it as the driving signal throughout training. This difference allows MLO to address known failure modes of modular pipelines, such as generating visually realistic but task-irrelevant data, or adapting examples unnecessarily.

### F.6. Separating Prompt Generation and Annotation in Multi-Level Optimization

In Section 4.2, it is possible to implement the PGM as an LLM, but in our framework, the PGM and annotation LLM need to be separate, as they serve different purposes and are optimized independently. The PGM generates task-specific prompts; the annotation LLM generates labels based on those prompts. The prompts generated by the PGM are used as input to the annotation LLM. This setup forms a three-level optimization: (1) the PGM is trained to generate prompts, (2) the annotation LLM produces labels and a downstream model is trained, and (3) validation loss is used to jointly update the PGM's hyperparameters and the annotation LLM's weights.

## G. Limitations

MLO generally incurs higher computational overhead than modular approaches due to nested optimization. However, several studies have shown that this cost can be reduced. For instance, Gala & Xie (2024) apply cost-reduction strategies such as reducing hyperparameter update frequency and using parameter tying, which brings MLO's runtime close to modular baselines. Similarly, Guo et al. (2025) show that although MLO has a higher per-epoch cost, it converges in fewer epochs, resulting in comparable total training time.

Regarding the method proposed in Section 5.2, a limitation is the reliance on simulation models. If a simulator fails to reflect target domain characteristics, generated data may be suboptimal. Simulator construction also requires domain expertise, which can hinder generalizability. In domains without simulators, pretrained generative models (e.g., diffusion models or LLMs) can synthesize candidate data or annotations. These generators, when properly conditioned, can replace simulators and be integrated into the MLO framework.

Moreover, our work focuses on data-scarce supervised settings, where limited labeled data exists. Our method assumes access to a small labeled dataset that can be split into training and validation sets. In fully unsupervised scenarios with no manual labels, our framework is not applicable.

