# OpenReview forum: "Position: Creating High-Fidelity Synthetic Training Data Should Employ Multi-level Optimization"
_ICML.cc/2026/Position_Paper_Track — ICML 2026 Position Paper Track regular_

### Official Review · Reviewer_ka9Z · 2026-03-11

**Significance:** 3
**Argument Clarity:** 2
**Rating:** 2
**Confidence:** 4

**Questions:**

NA

**Alternative Views Section:**

Yes

**Compliance With Llm Reviewing Policy A Conservative:**

Affirmed.

**Discussion Potential:**

2

**Final Justification:**

I stand by my opinion.

However, I noticed that the other three reviewers all gave it an "accept" rating, and I don't want my potentially flawed opinion to cause this potentially excellent paper to be rejected.

**I request that the AC disregard my review.**

**Paper Summary:**

This paper argues that creating high-fidelity synthetic training data should be approached through multilevel optimization (MLO) rather than through separately optimized pipelines.

The paper’s central position is that MLO provides a unified framework for coupling multi stages end-to-end so that downstream validation performance can guide upstream data decisions.

**Position:**

Yes

**Position In Title:**

Yes

**Related Work:**

3

**Strengths And Weaknesses:**

**Pro.**

This paper addresses a timely and relevant topic for the ICML community. Synthetic data, weak supervision, automatic labeling, domain adaptation, and data-centric optimization are all important themes, especially in low-resource or privacy-constrained settings.

The submission brings together a broad set of literatures under a common MLO lens, and the concrete optimization formulations make the discussion more useful than a purely abstract.

**Con.**

I do not think the current manuscript sufficiently supports the strength of its stated position.

1. The core claim is not merely that MLO is useful or promising, but that creating high-fidelity synthetic training data should employ MLO, and at several points the paper suggests that MLO is even necessary. This is a much stronger claim than what the paper actually substantiates.

2. The paper reads more like a broad survey than a sharply argued position paper. Much of the manuscript organizes prior work and sketches how various pipelines can be written in MLO form. That in itself can be useful, but the conceptual contribution beyond this literature organizing synthesis remains unclear.

As a suggestion, this submission should at least point out some viewpoints that seem to have been overlooked.

**Support:**

3

---

> ### Author Rebuttal · Authors · 2026-03-30
>
> We thank the reviewer for the constructive and valuable feedback, which we will leverage to improve this work.
> ### **1. Scope of the “Necessity” Claim**
>
> Thank you for this important comment. Our intention is not to assert that MLO is universally required, but rather that it provides a **principled and unifying framework** that is particularly well-suited for creating high-fidelity synthetic data when multiple interdependent stages must be coordinated.
>
> In the revision, we will **moderate the language** (e.g., replacing “necessary” with “well-suited” or “particularly effective”) and clearly position the paper as a **position and unifying perspective**, supported by illustrative case studies rather than exhaustive empirical validation.
>
> We will also **explicitly scope the claim** to settings where MLO is most beneficial—such as data-scarce domains, tasks with complex/structured outputs, and scenarios where data quality is the main bottleneck—while acknowledging that **simpler modular pipelines can be sufficient** when high-quality labeled data is abundant or when stages are weakly coupled.
>
> Finally, we will add a short discussion comparing MLO with alternative approaches (e.g., heuristic pipelines, iterative self-training), clarifying that MLO’s advantage lies in **end-to-end, validation-driven coordination across stages**, rather than being the only viable solution.
>
>
> $~$
>
>
> ### **2. Position vs. Survey and Conceptual Contribution**
>
> Thank you for this thoughtful comment. Our primary goal is not to survey existing work, but to argue that **multi-level optimization (MLO) provides a unifying principle for end-to-end data creation guided by downstream performance**, which is not explicitly formulated in prior literature. While we organize existing methods under the MLO perspective, this is intended to highlight a **previously implicit connection**—namely, that data generation, annotation, adaptation, and selection can be jointly optimized within a single framework rather than treated as disjoint modules.
>
> To better convey this contribution, we will revise the paper to:
> - **Sharpen the central thesis** early in the introduction and reduce survey-style exposition.
> - Clearly distinguish between **prior work** and our **proposed unifying perspective**, emphasizing what is new.
> - Add a concise subsection explicitly summarizing the **conceptual advances**, including: (i) framing data creation as a hierarchical optimization problem, (ii) enabling validation-driven feedback across stages, and (iii) unifying multiple paradigms (generation, annotation, adaptation, selection) under a single objective.
> - Streamline or move less critical literature discussions to improve focus.
>
> We will also clarify that the included case studies are **illustrative instantiations** of this perspective, rather than standalone contributions, reinforcing the paper’s role as a **position paper with a clear, focused argument**.
>
>
>
> $~$
>
>
>
> ### **3. Overlooked Alternative Viewpoints**
>
> Thank you for the helpful suggestion.
>
> In the revision, we will add a discussion of complementary perspectives, including:
> - **Model-centric approaches** (e.g., distillation, representation learning) that improve performance without modifying the data pipeline,
> - **Heuristic or modular pipelines**, which offer simplicity, reusability, and strong empirical performance in many settings,
> - **Iterative/self-training paradigms**, which provide lightweight forms of feedback without full end-to-end optimization.
>
> We will clarify that MLO is **not the only solution**, but a **complementary framework** that is particularly useful when strong coupling across stages and data quality are the main bottlenecks.

---

> > ### Author Rebuttal · Reviewer_ka9Z · 2026-04-02
> >
> > I stand by my opinion.
> >
> > However, I noticed that the other three reviewers all gave it an "accept" rating, and I don't want my potentially flawed opinion to cause this potentially excellent paper to be rejected.
> >
> > I request that the AC disregard my review.

---

### Official Review · Reviewer_8nma · 2026-03-12

**Significance:** 4
**Argument Clarity:** 3
**Rating:** 5
**Confidence:** 5

**Questions:**

Please refer to the Weaknesses. If the authors can address these concerns, I would be willing to increase my score.

**Alternative Views Section:**

Yes

**Compliance With Llm Reviewing Policy A Conservative:**

Affirmed.

**Discussion Potential:**

3

**Final Justification:**

The rebuttal has satisfactorily addressed my concerns.

**Paper Summary:**

This paper argues that data generation, annotation, adaptation, and selection should not be treated as isolated data processing modules. Instead, they should be formulated as a hierarchical optimization problem driven by downstream performance. Through a series of examples from data generation, automatic labeling, source domain adaptation, and prompt learning, the paper shows that multi-level optimization can serve as a unified perspective across these tasks.

**Position:**

Yes

**Position In Title:**

Yes

**Related Work:**

2

**Strengths And Weaknesses:**

**Strengths:**

1. The problem discussed in this paper is highly important. A key question is how to leverage existing models to create a data flywheel that continuously improves models through iterative data generation and refinement.

2. The paper is well organized. It places several data-centric problems, which are often discussed separately, under a unified hierarchical optimization perspective. It improves conceptual coherence and helps to see the shared structure across these problems. Upstream decisions affect downstream models, while downstream validation signals in turn constrain the upstream process.

3. The paper goes beyond general advocacy. It analyzes several concrete scenarios, including complex label generation, self-consistency verification for automatic annotation, category-performance-driven data generation, and sample-level source data adaptation. For each case, the paper explicitly describes the hierarchical structure, the roles of variables, and the direction of optimization. The position is therefore supported by structured reasoning rather than purely conceptual arguments.

**Weaknesses:**

1. The paper emphasizes an end-to-end optimization framework. In contrast, modular optimization frameworks often offer greater reusability across tasks. It is unclear whether this new perspective is practically feasible.

2. The argument relies on a strong assumption. The complex label generation stage constructs a dataset for training a data generation model by swapping input and output pairs and assumes a one-to-one mapping. This assumption may hold in some structured prediction tasks, but it is less natural in more general vision or language tasks.

3. Some recent distillation methods bypass data generation and directly enhance the model. Recent excellent studies on generation enhanced understanding [1,2,3,4] follow this direction and distill knowledge from the internal representations of a model to achieve model improvement without labels. This approach also forms an end-to-end optimization framework. The paper should clarify its potential advantages over these methods or explain whether such approaches can be incorporated into its perspective.

[1] Diffusion Feedback Helps CLIP See Better. ICLR 2025.

[2] GenHancer: Imperfect Generative Models are Secretly Strong Vision-Centric Enhancers. ICCV 2025.

[3] un2CLIP: Improving CLIP's Visual Detail Capturing Ability via Inverting unCLIP. NeurIPS 2025.

[4] Guiding Diffusion-based Reconstruction with Contrastive Signals for Balanced Visual Representation. CVPR 2026.

**Support:**

2

---

> ### Author Rebuttal · Authors · 2026-03-30
>
> We thank the reviewer for the constructive and valuable feedback, which we will leverage to improve this work.
>
>
> ### **1. End-to-End vs. Modular Frameworks and Practical Feasibility**
>
> Thank you for this insightful comment. We agree that modular frameworks offer advantages in reusability and simplicity, and our intention is not to replace them universally, but to highlight complementary benefits of end-to-end optimization.
>
> First, we will clarify that the proposed MLO framework can be implemented in a **hybrid manner**, where existing modular components (e.g., generators, annotators, downstream models) are reused, and MLO is applied to coordinate a subset of stages. This preserves much of the reusability of modular designs while enabling validation-driven feedback across components.
>
> Second, regarding feasibility, we will expand the discussion to emphasize that MLO does not require fully unrolled, large-scale joint optimization. In practice, **approximate and efficient strategies** (e.g., alternating updates, truncated optimization, and partial differentiation) can make the framework tractable. We will also highlight that bi-level optimization, a special case of MLO, has already been successfully applied in several domains, suggesting practical viability.
>
> Third, we will clarify that the benefits of MLO are most pronounced in settings where **interactions between stages are strong** and modular pipelines struggle to align data with downstream objectives. In such cases, the additional coordination enabled by MLO can outweigh the loss in modular simplicity.
>
> Finally, we will revise the manuscript to better position MLO as a **complementary framework** that can augment, rather than replace, modular approaches, and provide clearer guidance on practical implementation strategies.
>
>
>
>
> $~$
>
>
> ### **2. One-to-One Mapping Assumption in Data Generation**
>
> Thank you for this important observation. We would like to clarify that this assumption is introduced **for simplicity in the illustrative example**, rather than being a requirement of the proposed MLO framework. The core idea of MLO does not depend on one-to-one mappings. In more general settings, the data generation model can be formulated as learning **conditional or stochastic mappings**, which naturally capture one-to-many relationships.
>
> In the revision, we will explicitly clarify this point and extend the discussion to more general formulations, including:
> - Conditional generative models that produce diverse outputs for a given label,
> - Latent-variable models that capture multi-modal mappings,
> - Joint modeling approaches that avoid assuming invertibility between inputs and labels.
>
> We will also emphasize that the swapping construction is one possible instantiation for structured tasks, but not a limitation of the overall framework.
>
>
>
> $~$
>
>
>
> ### **3. Relation to Distillation and Generation-Enhanced Understanding**
>
>
> Thank you for this insightful comment and for pointing out these relevant works. We would like to clarify that these approaches are **complementary rather than conflicting** with our perspective. While they focus on improving model representations (e.g., via distillation or reconstruction), our framework focuses on **optimizing the data creation process** (generation, annotation, adaptation, selection) to better align with downstream tasks. These represent two orthogonal axes: *model-centric optimization* vs. *data-centric optimization*.
>
>
> Importantly, such methods can be naturally **incorporated into the MLO framework**. For example, distillation or reconstruction objectives can serve as part of the upper-level or intermediate objectives, providing additional feedback signals beyond standard validation loss. In this sense, MLO can unify both paradigms by jointly optimizing data and model components in an end-to-end manner.
>
>
> In terms of advantages, MLO is particularly beneficial in scenarios where **data quality and alignment with task objectives are the primary bottlenecks**, whereas distillation-based methods focus on transferring or refining representations within a fixed data pipeline. MLO explicitly optimizes the *input distribution* seen by the model, which can be critical in data-scarce or distribution-shift settings.
>
>
> In the revision, we will add a discussion comparing these approaches and clarifying how they relate to and can be integrated into the proposed framework.

---

> > ### Author Rebuttal · Reviewer_8nma · 2026-04-01
> >
> > I have carefully reviewed the authors' responses to my comments, together with the feedback from the other reviewers. The rebuttal has satisfactorily addressed my concerns, and I will therefore increase my score to Accept.

---

### Official Review · Reviewer_kwJE · 2026-03-12

**Significance:** 2
**Argument Clarity:** 3
**Rating:** 5
**Confidence:** 3

**Questions:**

Is computational complexity the only drawback/limitation of using multi-layer optimization? Are there potential other risks, e.g. shortcuts or biases?

How model-dependent are the generated data/labels? Can they be reused to train other (similar) models? If so, what could be potential tradeoffs?

**Alternative Views Section:**

Yes

**Compliance With Llm Reviewing Policy A Conservative:**

Affirmed.

**Discussion Potential:**

2

**Final Justification:**

The authors' rebuttal addresses my concerns.

**Paper Summary:**

The paper argues for multi-level optimization to better address the challenges of creating high-quality synthetic data to address data scarcity in model training. Often, such data generation pipelines consist of multiple un- or loosely coupled steps, which the paper argues leads to sub-optimal results, proposing to couple these steps as a multi-level optimization problem. In this regard, the paper discusses a series of possible use cases ranging from end-to-end training data generation to end-to-end annotation of unlabeled data and end-to-end sample adaptation and selection.

**Position:**

Yes

**Position In Title:**

Yes

**Related Work:**

3

**Strengths And Weaknesses:**

Strengths:
- [Good fit for MLO] Well motivates the fit for multi-level optimization to improve the quality of the generated data//labels/selections.

- [Nicely describes multiple use cases] The examples where multilevel optimization might be useful encompass several examples with a high-level description on how it could be used.

Weaknesses:
- [Risks of MLO] While the paper well argues the potential benefits of multilevel optimization in several scenarios, it only marginally discusses potential drawbacks (mainly limited to Section 7 and the computational complexity of the method). It is unclear what other drawbacks there might be, e.g., is there a risk for multi-layer optimization to introduce bias or to find undesired shortcuts?

- [Computational complexity] The drawback of the additional complexity should be highlighted earlier in the paper, and possibly provide some insights on its quantification. For example, to obtain the results shown in Figure 1

- [Lower/no reuse opportunity] One advantage of generated data is that it can potentially be used across multiple different model trainings to offset the costs incurred in generating high-quality data. The proposed approach seems to potentially disrupt this "economy of scale".

**Support:**

3

---

> ### Author Rebuttal · Authors · 2026-03-30
>
> We thank the reviewer for the constructive and valuable feedback, which we will leverage to improve this work.
>
>
> ### **1. Risks and Potential Drawbacks of MLO**
>
> Thank you for this important comment.  Beyond computational complexity, we acknowledge several additional potential drawbacks:
>
> **(1) Risk of bias amplification.**
> Since MLO optimizes data using downstream performance, it may amplify biases in the validation signal or model, especially with unbalanced or imperfect data, leading to biased synthetic data.
>
> **(2) Shortcut learning and overfitting to signals.**
> The framework may exploit spurious correlations that improve validation performance without true generalization, especially with narrow or noisy signals.
>
> **(3) Optimization instability.**
> Multi-level optimization introduces nested dependencies, making training more sensitive to initialization, hyperparameters, and update schedules.
>
> **(4) Reduced interpretability.**
> End-to-end optimization may make it harder to disentangle individual component contributions compared to modular pipelines.
>
> We will clarify mitigation strategies, including diverse or cross-validated validation signals, robustness objectives (e.g., worst-case performance), and regularization or diversity constraints to prevent collapse.
>
> We will expand Section 7 to include these points, providing a more balanced discussion of both the benefits and risks of MLO.
>
> $~$
>
> ### **2. Computational Complexity and Early Discussion**
>
> Thank you for this helpful suggestion. We agree that the computational complexity of MLO should be highlighted earlier and quantified more clearly.
>
> In the revision, we will move this discussion to an earlier section and provide explicit **complexity analysis**, including scaling with the number of levels, optimization steps, and model components. We will also clarify how techniques such as truncated optimization and alternating updates reduce practical cost.
>
> The table below reports the GPU hours required to obtain the skin lesion segmentation results on ISIC shown in Figure 1.
>
> | Method     | Runtime (GPU hours) |
> |------------|---------------------|
> | Vanilla    | 0.5                 |
> | Rotate     | 0.9                 |
> | Flip       | 0.9                 |
> | Translate  | 0.9                 |
> | Combine    | 1.1                 |
> | WGAN       | 1.2                 |
> | GenSeg     | 1.5                 |
>
> We will include these compute times in the revised paper, along with similar measurements for the other segmentation tasks.
>
>
> $~$
>
>
>
> ### **3. Reusability and “Economy of Scale”**
>
> Thank you for this insightful comment. MLO does not preclude reuse but enables a **reuse–specialization tradeoff**. In practice, components such as data generation or annotation can be reused across models, while higher-level adaptation or selection is fine-tuned for specific tasks, preserving the “economy of scale.”
>
> MLO can also produce **higher-quality, more generalizable data**, potentially improving transferability across models compared to heuristic pipelines.
>
> In addition, MLO can be applied **selectively or periodically** (e.g., refining a shared dataset), rather than requiring full end-to-end optimization for every model, allowing a balance between reuse and task-specific optimization.
>
> We will clarify these points and associated tradeoffs in the revision.
>
>
>
> $~$
>
>
>
> ### **4. Model Dependence and Reusability of Generated Data**
>
>
> Thank you for this important question. The model dependence of generated data in MLO lies on a spectrum.
>
>
> Because MLO optimizes data using downstream performance, the generated data can be **partially model-dependent**, especially when tightly coupled to a specific architecture or validation signal. However, it can still be **reusable across similar models**, particularly when models share inductive biases or when optimization incorporates multiple models, validation signals, or emphasizes robustness and diversity.
>
>
> There is a natural **tradeoff between specialization and generality**:
> - **Higher specialization** improves performance for a specific model but reduces transferability.
> - **Higher generality** improves reuse but may sacrifice peak performance.
>
>
> This tradeoff can be controlled by using model ensembles, diverse validation criteria, or decoupling components (e.g., reusing generators while adapting selection strategies).
>
>
> Overall, MLO introduces a tunable balance between **task-specific optimization and cross-model transferability**, rather than eliminating reusability.

---

> > ### Author Rebuttal · Reviewer_kwJE · 2026-04-02
> >
> > The rebuttal nicely extends the discussion to a more rounded, complete perspective. I hence increased my score.

---

### Official Review · Reviewer_CssW · 2026-03-12

**Significance:** 3
**Argument Clarity:** 3
**Rating:** 5
**Confidence:** 4

**Questions:**

1. The framework seems to assume a validation set, what if this validation set is small or it is not available in the beginning?
2. Since the proposed framework is more complex than the modular design, are there any potential measurement to help the researcher/user to decide when to adopt this framework? Or say in which case this framework will be more beneficial?

**Alternative Views Section:**

Yes

**Compliance With Llm Reviewing Policy A Conservative:**

Affirmed.

**Discussion Potential:**

3

**Paper Summary:**

This position paper argues that multi-level optimization (MLO) should be considered for creating high-fidelity synthetic training data. The point is that existing pipelines typically optimize generation / annotation / adaptation / selection largely in isolation and therefore fail to systematically use downstream model performance to guide upstream data generation. The paper presents MLO as a unifying way to connect these stages end-to-end using nested optimization problems.

**Position:**

Yes

**Position In Title:**

Yes

**Related Work:**

3

**Strengths And Weaknesses:**

Stength
1. The paper identifies a failure mode that objective misalignment in modular pipelines from generation, annotation and domain adaptation and proposes MLO as a principled framework to connect data generation with downstream validation performance. Rather than focusing solely on generation, the paper proposes to use a end-to-end framework that spans from generation to hyperparameter selection.
2. Empirical evidence aligns with the position. The GenSeg case study claims consistent gains over augmentation and generator baselines on multiple medical segmentation datasets.

Weakness
1. It will be better to have more detailed complexity and benefit tradeoff. Currently the paper points out some potential feasibility, but a more structured discussion of compute/memory/tuning burden will be helpful.
2. By tightly optimizing synthetic data generation and annotation against a particular downstream model and validation split, the approach risks producing data that is overly specialized and thus lead to easy overfitting.

**Support:**

3

---

> ### Author Rebuttal · Authors · 2026-03-30
>
> We thank the reviewer for the positive and valuable feedback, which we will leverage to improve this work.
>
>
> ### **1. Complexity–Benefit Tradeoff**
>
> Thank you for this valuable suggestion. In the revision, we will add a subsection analyzing compute cost, memory usage, and tuning burden, alongside benefits.
>
> **(1) Computational complexity.**
> MLO introduces overhead from nested optimization and feedback propagation, potentially scaling with the number of levels. However, techniques such as truncated optimization, implicit differentiation, and alternating updates can reduce this cost. The added compute improves data quality, which can lower downstream training iterations and data requirements.
>
> **(2) Memory footprint.**
> Naive implementations can be memory-intensive due to unrolled optimization. We will discuss efficient strategies (e.g., checkpointing, implicit gradients, stage-wise approximation) that avoid full unrolling.
>
> **(3) Tuning and implementation burden.**
> MLO introduces additional hyperparameters, but many components align with existing pipelines and can be reused. It also reduces reliance on manual heuristics by replacing them with optimization-driven procedures.
>
> **(4) Benefit–cost tradeoff.**
> MLO is most beneficial in data-scarce or high-cost domains, tasks with complex/structured outputs, and scenarios where data quality is the main bottleneck. In these cases, better alignment with downstream objectives can improve sample efficiency, robustness, and reduce dependence on large labeled datasets.
>
> $~$
>
>
> ### **2. Overfitting Risk from Tight Coupling to Downstream Model**
>
> Thank you for this important comment. The proposed MLO framework is not inherently tied to a single model or validation split. In practice, overfitting can be mitigated by using multiple validation splits (e.g., cross-validation) and multiple model architectures or initializations to provide more robust feedback signals.
>
> Regularization can also be integrated at different levels of MLO, such as encouraging diversity in data generation, along with techniques like early stopping and stochastic optimization to reduce overfitting.
>
> In addition, MLO can incorporate robustness objectives (e.g., worst-case or class-specific performance), which promote generalization rather than fitting to specific validation examples.
>
> Overall, MLO aims to align synthetic data with downstream task requirements, not to memorize a validation set. Compared to heuristic pipelines, it provides a principled way to balance data fidelity and generalization. We will clarify these points and expand discussion of overfitting mitigation in the revision.
>
>
> $~$
>
>
>
> ### **3. Availability and Size of Validation Set**
>
> Thank you for this important question. The framework does not require a large held-out validation set. In data-scarce settings, strategies such as cross-validation (e.g., k-fold) can provide reliable validation signals from limited labeled data and integrate naturally into MLO.
>
> When labeled validation data is unavailable initially, proxy objectives can be used, such as self-supervised or unsupervised criteria (e.g., likelihood, consistency), model agreement, or pseudo-label confidence. As labeled data becomes available, the framework can transition to true validation loss.
>
> Moreover, MLO is flexible in its upper-level objectives and can incorporate multiple supervision sources, including small labeled sets, weak supervision, or domain-specific constraints, reducing reliance on a single validation set.
>
> We will clarify that “validation signal” in MLO can take different forms depending on data availability and add discussion on practical strategies for low-resource settings.
>
>
> $~$
>
>
>
> ### **4. When to Adopt the MLO Framework**
>
> Thank you for this insightful question. We will add a structured discussion outlining **practical criteria** for adopting MLO. The framework is most beneficial when **data quality is the primary bottleneck** rather than model capacity, such as in (i) data-scarce domains (e.g., healthcare), (ii) tasks with **complex or structured outputs** (e.g., segmentation, molecular properties), and (iii) scenarios with **distribution mismatch** requiring adaptation or selection. In these settings, aligning data with downstream objectives can yield greater gains than model scaling.
>
> We will also clarify **measurable indicators** to guide adoption, including large train–validation gaps, class imbalance or class-specific failures, and strong sensitivity to data curation strategies.
>
> In addition, MLO can be introduced **incrementally**, e.g., starting with bi-level optimization on a single component and expanding if beneficial, allowing controlled tradeoffs between complexity and gain.
>
> Finally, we will emphasize that MLO is not always necessary. When high-quality labeled data is abundant and models generalize well, simpler pipelines may suffice. These guidelines will be added to improve practical clarity.

---

> > ### Author Rebuttal · Reviewer_CssW · 2026-04-03
> >
> > My concerns are fully resolved. Thanks for the clarification!

---

### Decision · Program_Chairs · 2026-04-30

**Decision:**

Accept (regular)

**Comment:**

The reviews consistently acknowledge the paper’s contribution and its positioning. There was one negative score, however, the reviewer seem to self-disregard their score which makes it unanimously acceptance